# Vibration Characteristics of a Laminated Composite Double-Cylindrical Shell System Coupled with a Variable Number of Annular Plates

**DOI:** 10.3390/ma15124246

**Published:** 2022-06-15

**Authors:** Ying Zhang, Dongyan Shi, Dongze He

**Affiliations:** 1College of Mechanical and Electrical Engineering, Harbin Engineering University, Harbin 150001, China; zhangying2016@hrbeu.edu.cn (Y.Z.); hdz2012071506@126.com (D.H.); 2College of Engineering, Heilongjiang Bayi Agricultural University, Daqing 163319, China

**Keywords:** laminated composite double-cylindrical shell system, improved Fourier series, Rayleigh–Ritz method, free vibration, steady-state response

## Abstract

A vibration characteristic analysis model of a laminated composite double cylindrical shell system (LCDCSS) coupled with several annular plates under general boundary conditions is established. Artificial springs are used to simulate the coupling relationship between substructures to ensure the continuity of displacement both at ends of the shells and coupling boundaries. The variable number of annular plates can be distributed unevenly and coupled elastically. Displacement functions of LCDCSS are expressed with improved Fourier series. Based on the principle of energy, obtain the unknown coefficients of the displacement components by using the Rayleigh–Ritz method. The convergence and effectiveness of the proposed method are verified by comparing with the results with literature and FEM, and then carried out parametric investigation to study the free and steady-state response vibration characteristics of LCDCSS. Rapid prediction of free vibration and response vibration of a double-layer cylindrical shell system with various structures and scales is realized by exploiting the model, and some new results of double-layer cylindrical shell system are explored, which can provide reference for further research.

## 1. Introduction

As a basic structural member, laminated composite cylindrical shells are used in aviation equipment, ship engineering, construction, transportation, chemical engineering, and other engineering fields. In addition, a double-layer cylindrical shell has been more and more widely used in various fields for excellent physical and chemical properties with the technology progress, such as a typical structure of submarine cabin, seabed oil pipeline, and so on. Studies on vibration characteristics of a double cylindrical shell system have important theoretical significance and practical application value. Scholars continue to explore theoretical and experimental methods for solving the dynamic problems of various kinds of cylindrical shell structures to predict and control the vibration of structures. In recent decades, a number of research studies have been carried out around laminated cylindrical shell structures, which are fully recorded in the literature.

Ming et al. [1] proposed a model to measure the structural characteristics of cylindrical shells, in which the point force excitation is used to replace the circumferential modal force that is difficult to realize in practice. In this method, the transfer function components of different circumferential modes are obtained from the measured data by using the least square method, and the feasibility of this method is verified by the point force excitation experiments of cylindrical shells. Lee and Reddy [2] used the higher-order shear deformation theory to study the vibration characteristics of laminated shell structures and search for a way to control it. Based on the classical shell theory (CST), Zhong et al. [3] proposed a new exact solution to analyze free vibration of cross-ply laminated composite cylindrical shells. Hu et al. [4] studied vibration frequency of a laminated cylindrical thin panel with the Rayleigh–Ritz method, which combined the principle of virtual work with the displacement function of two-dimensional algebraic polynomials, the vibration control equations of laminated cylindrical thin plates with torsion and curvature are established. Based on the first order shear deformation theory, Qu et al. [5] proposed a unified formulation for vibration analysis of composite laminated shell considered both the effects of shear distortion and rotary inertia. Maharjan et al. [6] evaluated the elastic properties of laminated composite cylindrical shells using micro-mechanical approaches. Hafizah et al. [7] analyzed vibration of antisymmetric angle-ply composite annular plates with variable thickness. Civalek [8] presented vibration analysis of laminated composite conical shells based on the shear deformation theory. Zhao et al. [9] proposed a unified analysis model to present the free vibration of laminated composite elliptic cylinders under various boundary conditions.He et al. [10] analyzed the free vibration of composite laminated cylindrical shells with general boundary conditions using a wave-based method (WBM). Kumar [11] explored vibration of laminated composite skew hypar shells by using higher order theory. Jin et al. [12,13,14] put forward a unified improved Fourier solution for composite laminated structural elements with arbitrary elastic constraints, which can be used to solve the free vibration problems of cylinders, cones, spherical shells and annular plates. Wang et al. [15,16,17] offered unified solutions for dynamic analysis of circular, annular and sector plate structures of orthotropic materials, laminated composites and functionally graded materials under general boundary conditions. Tornabene et al. [18] completed a general higher-order equivalent single layer theory to study free vibrations of doubly-curved laminated composite shells and panels. Li et al. [19,20] analyzed the vibration characteristics of rotating composite laminated cylindrical shells under subsonic air flow and axial load in humid and hot environments. Zhang et al. [21] analyzed vibration of a composite laminated sector, annular, and circular plate with a simplified plate theory. Zuo et al. [22] combined general shell theory and the wavelet finite element method to present static and vibration characteristic of laminated composite shells. Liu et al. [23] studied the free vibration characteristics of functionally graded cylindrical shells by using the wave-based method. Bisheh et al. [24] carried out free vibration analysis of piezoelectric coupled carbon nanotube (CNT) reinforced composite cylindrical shells, and discussed the influence of boundary conditions on the frequency.

With the development of the research and the expansion of engineering, various complex coupled structures based on cylindrical shells have attracted widespread attention. Some researchers have paid attention to the double shell system. Yamada et al. [25] obtained the vibration control equations of a cylindrical shell using the transfer matrix, and presented the free vibration of a circular cylindrical double-shell system. Yuan et al. [26] established the free vibration characteristic analysis model of a ring cylindrical shell coupling structure system by using the Rayleigh–Ritz method, in which the coupling connection conditions of substructure are simulated by artificial spring technology. Jin et al. [27] explored vibration analysis of circular cylindrical double-shell structures under general coupling and end boundary conditions. Chen et al. [28] derived the dynamic equilibrium equation of the double elastic spherical shell, and obtained the semi analytical solutions of vibration and sound radiation of underwater double spherical shell by using the Dirac-delta function. Dogan et al. [29] integrated the nonlinear spring damper model into the system, established the analytical model of a nonlinear response of a double wall sandwich cylindrical shell system under random excitation, and gave the nonlinear response law. Qing et al. [30] proposed a hybrid state variable technique, which extended the semi-analytical method to the natural frequency and mode analysis of double-layer thick shell structures. Wang et al. [31] carried out a research project on dynamic failure behaviors of steel double-layer latticed cylindrical shell. Zhang et al. [32] presented vibration and sound radiation from submerged double cylindrical shells using the modal superposition method. Huang et al. [33] obtained the wind effects on the double-layer cylindrical latticed shell (DCLS). Zhang et al. [34] studied the free vibration analysis of double cylindrical shells that is rib stiffened, based on a modified Fourier-Ritz method. Xie et al. [35] obtained vibration analysis of double-walled cylindrical shells using a wave-based method. Wali et al. [36] studied free vibration analysis of FGM shell structures by building an efficient 3D-shell model. Choi et al. [37] presented the free vibration of double cylindrical shells based on transfer of an influence coefficient. Mehdi et al. [38] researched vibration of double-bonded micro sandwich cylindrical shells under multi-physical loadings. Chen et al. [39] obtained a vibration frequency model of a ring stiffened cylindrical shell that was stiffened with intermediate ribs by using a wave-based method.

From literature review, researchers have been widely exploring the vibration characteristics of cylindrical shell structures. However, it is regrettable that study of laminated composite double cylindrical shells is seldom done. There may be some limitations to predict vibration character of a double cylindrical shell system with the coupling relationship, scale or boundary conditions of the system changed. The research on predicting the vibration characteristics of composite double cylindrical shell system coupled with variable annular plates under general boundary conditions and coupling constraints has not been reported.

In this research, a unified analytical model for the vibration characteristics of laminated double cylindrical shells coupling with a variable number of annular plates is established. Artificial virtual boundary technique and virtual coupling spring technique are used to simulate the constraint relationship. Based on the first order shear deformation shell theory, effects of tension, bending, and torsion are taken care of. Displacement admissible functions of the system were approached by improved Fourier series. According to the Rayleigh–Ritz method, the unknown coefficients of displacement components were obtained based on the principle of energy. After verifying the convergence and correctness of the proposed method by numerical examples, a series of parametric studies are analyzed to predict the free and state response vibration characteristics of LCDCSS.

## 2. Analysis Model of the Laminated Composite Double Cylindrical Shell Structure

### 2.1. Description of the Model

Figure 1 describes the structure of a laminated composite double cylindrical shell system coupled with general boundary conditions. There are two coaxial cylinders with the same length are connected with a variable number of annular plates which distributed non-uniformly along the axial direction. For the convenience of model description, cylindrical coordinates (*o*, *x*, θ, *r*) are set up along the axial, circumferential, and radial direction of the shell structure. The geometric properties of all cylindrical shell structure and annular plates are described based on the cylindrical coordinate system. Length, radius, circumferential angle, and thickness of shells are expressed by *L*, *R_p_*, *h_p_*, θ*_p_*, in which *p* = *i,o* represent the inner and outer shell individually (Cylinder 1 and 2 in Figure 1). In a similar way, the structure of annular plates described with the cylindrical coordinate system (*o*, *r*, θ, *z*), *b* = *R_o_* − *R_i_* is the width of the annular plate, which is the direction from the internal to outer edge along the radial direction. *h_c_* represents the thickness of annular plate. *u_g_*, *v_g_*, *w_g_* (*g* = *c*, *a*) stand for middle-layer displacements of laminated cylindrical shells and annular plates, respectively. $p_{a}^{q}$ denotes position of the *q*th annular plate along the axial direction of the cylindrical shell.

### 2.2. Kinematic Relations and Stress Resultants

#### 2.2.1. For Cylindrical Shell

Based on shear shell theory (SDST), both in-plane and out-of-plane vibration are considered to describe the stress–strain relationship of a cylindrical shell structure. The displacement and rotation components of any point on the cylindrical shell structure can be represented as follows [12]:(1)$$\begin{array}{l} U^{p}\left(x,\theta ,z,t\right)=u_{c}^{p}\left(x,\theta ,t\right)+z\phi_{cx}^{p}\left(x,\theta ,t\right)\\ V^{p}\left(x,\theta ,z,t\right)=v_{c}^{p}\left(x,\theta ,t\right)+z\phi_{c\theta }^{p}\left(x,\theta ,t\right)\\ W^{p}\left(x,\theta ,z,t\right)=w_{c}^{p}\left(x,\theta ,t\right) \end{array}$$
where *p* = *i,o* stand for inner and outer cylindrical shells in the system. *t* is time variable. *U^p^*, *V^p^*, and *W^p^* represent displacement for an arbitrary point of the cylindrical shell, in axial, circumferential, and radial direction. $u_{c}^{p}$*, $v_{c}^{p}$,*
$w_{c}^{p}$ are middle layer displacement of the cylindrical shell, and $\varphi_{cx}^{p}$*, $\varphi_{c\theta }^{p}$,* present rotatory displacement component along *θz* and *xz* surfaces, respectively. According to the linear elasticity theory [12], strain and displacement relationships are as follows:(2)$$\varepsilon_{xx}^{p}=\varepsilon_{xx0}^{p}+z\chi_{xx}^{p},\;\varepsilon_{\theta \theta }^{p}=\varepsilon_{\theta \theta 0}^{p}+z\chi_{\theta \theta }^{p},\gamma_{x\theta }^{p}=\gamma_{x\theta 0}^{p}+z\chi_{x\theta }^{p},\;\gamma_{xz0}^{p}=\frac{\partial w_{c}^{p}}{\partial x}+\phi_{cx}^{p},\gamma_{\theta z0}^{p}=\frac{\partial w_{c}^{p}}{R_{p}\partial \theta }-\frac{v_{c}^{p}}{R_{p}}+\phi_{c\theta }^{p}$$
where $\varepsilon_{xx0}^{p}$, $\varepsilon_{\theta \theta 0}^{p}$, $\gamma_{x\theta 0}^{p}$ denote the structural strains in the middle surface, and $\chi_{xx}^{p}$, $\chi_{\theta \theta }^{p}$, $\chi_{x\theta }^{p}$ are mid-surface changes in curvature. The relationship with displacement can be shown as: (3)$$\varepsilon_{xx0}^{p}=\frac{\partial u_{c}^{p}}{\partial x},\;\varepsilon_{\theta \theta 0}^{p}=\frac{\partial v_{c}^{p}}{R_{p}\partial \theta }+\frac{w_{c}^{p}}{R_{p}},\gamma_{x\theta 0}^{p}=\frac{\partial v_{c}^{p}}{\partial x}+\frac{\partial u_{c}^{p}}{R_{p}\partial \theta }\;\chi_{xx}^{p}=\frac{\partial \phi_{cx}^{p}}{\partial x},\chi_{\theta \theta }^{p}=\frac{\partial \phi_{c\theta }^{p}}{R_{p}\partial \theta },\chi_{x\theta }^{p}=\frac{\partial \phi_{cx}^{p}}{R_{p}\partial \theta }+\frac{\partial \phi_{c\theta }^{p}}{\partial x}$$

The thickness parameter of a fiber layer is between z_k_ < z < z_k + 1_; thus, the corresponding stresses are obtained in terms of the general Hooke’s law as:(4)$$\left\{\begin{array}{c} \sigma_{cxx}^{p}\\ \begin{array}{l} \sigma_{c\theta \theta }^{p}\\ \tau_{cx\theta }^{p} \end{array} \end{array}\right\}=\left[\begin{array}{ccc} {\bar{Q}}_{11}^{k} & {\bar{Q}}_{12}^{k} & {\bar{Q}}_{16}^{k}\\ {\bar{Q}}_{12}^{k} & {\bar{Q}}_{22}^{k} & {\bar{Q}}_{26}^{k}\\ {\bar{Q}}_{16}^{k} & {\bar{Q}}_{26}^{k} & {\bar{Q}}_{66}^{k} \end{array}\right]\left\{\begin{array}{c} \varepsilon_{xx0}^{p}\\ \begin{array}{l} \varepsilon_{\theta \theta 0}^{p}\\ \gamma_{x\theta 0}^{p} \end{array} \end{array}\right\},\left\{\begin{array}{c} \tau_{cxz}^{p}\\ \tau_{c\theta z}^{p} \end{array}\right\}=\left[\begin{array}{cc} {\bar{Q}}_{55}^{k} & {\bar{Q}}_{45}^{k}\\ {\bar{Q}}_{45}^{k} & {\bar{Q}}_{44}^{k} \end{array}\right]\left\{\begin{array}{c} \gamma_{xz}^{p}\\ \gamma_{\theta z}^{p} \end{array}\right\}$$
where $\sigma_{cxx}^{p}$, $\sigma_{c\theta \theta }^{p}$ are normal stresses, and $\tau_{cx\theta }^{p}$$\tau_{cxz}^{p}$ and $\tau_{c\theta z}^{p}$ are shear stresses [40]. The *k*th layer stiffness coefficients of the cylindrical shell are ${\bar{Q}}_{ij}$ (*i,j* = 1,2,4,5,6), which represent the elastic properties of the material of the layer, and can be determined as [17]:(5)$$\begin{array}{l} \left[\begin{array}{ccccc} {\bar{Q}}_{11}^{k} & {\bar{Q}}_{12}^{k} & {\bar{Q}}_{16}^{k} & 0 & 0\\ {\bar{Q}}_{12}^{k} & {\bar{Q}}_{22}^{k} & {\bar{Q}}_{26}^{k} & 0 & 0\\ {\bar{Q}}_{16}^{k} & {\bar{Q}}_{26}^{k} & {\bar{Q}}_{66}^{k} & 0 & 0\\ 0 & 0 & 0 & {\bar{Q}}_{44}^{k} & {\bar{Q}}_{45}^{k}\\ 0 & 0 & 0 & {\bar{Q}}_{45}^{k} & {\bar{Q}}_{55}^{k} \end{array}\right]=T\left[\begin{array}{ccccc} Q_{11}^{k} & Q_{12}^{k} & 0 & 0 & 0\\ Q_{12}^{k} & Q_{22}^{k} & 0 & 0 & 0\\ 0 & 0 & Q_{44}^{k} & 0 & 0\\ 0 & 0 & 0 & Q_{55}^{k} & 0\\ 0 & 0 & 0 & 0 & Q_{66}^{k} \end{array}\right]T^{T}\\ T=\left[\begin{array}{ccccc} c^{2} & s^{2} & 0 & 0 & -2sc\\ s^{2} & c^{2} & 0 & 0 & 2sc\\ 0 & 0 & c & s & 0\\ 0 & 0 & -s & c & 0\\ sc & -sc & 0 & 0 & c^{2}-s^{2} \end{array}\right],s=\sin\alpha_{fiber}^{k},c=\cos\alpha_{fiber}^{k}\\ Q_{11}^{k}=\frac{E_{1}^{k}}{1-\mu_{12}^{k}\mu_{21}^{k}},\;Q_{12}^{k}=\mu_{21}^{k}Q_{11}^{k},\;Q_{22}^{k}=\frac{E_{2}^{k}}{1-\mu_{12}^{k}\mu_{21}^{k}},\;Q_{44}^{k}=G_{23}^{k}\\ Q_{55}^{k}=G_{13}^{k},\;Q_{66}^{k}=G_{12}^{k} \end{array}$$
where $\alpha_{fiber}^{k}$ denotes the included angle between the *x*-axis and the principal direction of the *k*th layer. $Q_{ij}^{k}$ (*i*,*j* = 1,2,4,5,6) represents the laminated stiffness coefficients. $E_{1}^{k}$ and $E_{2}^{k}$ express the longitudinal modulus and transverse modulus of the *k*-th layer, $\mu_{12}^{k}$ and $\mu_{21}^{k}$ are the Poisson’s ratios. $G_{13}^{k}$, $G_{23}^{k}$, and $G_{12}^{k}$ are the shear moduli and $E_{1}^{k}$ = $E_{2}^{k}$,$G_{12}^{k}=G_{13}^{k}=G_{23}^{k}=E_{1}^{k}/2(1+\mu_{12})$.

By integrating the stresses along thickness, the force and moment resultants of laminated thick shells can be obtained as:(6)$$\left\{\begin{array}{c} \begin{array}{c} N_{cxx}^{}\\ N_{c\theta \theta }^{} \end{array}\\ N_{cx\theta }^{}\\ M_{cxx}^{}\\ M_{c\theta \theta }^{}\\ M_{cx\theta }^{} \end{array}\right\}=\left[\begin{array}{cccccc} A_{11} & A_{12} & A_{16} & B_{11} & B_{12} & B_{16}\\ A_{12} & A_{22} & A_{26} & B_{12} & B_{22} & B_{26}\\ A_{16} & A_{26} & A_{66} & B_{16} & B_{26} & B_{66}\\ B_{11} & B_{12} & B_{16} & D_{11} & D_{12} & D_{16}\\ B_{12} & B_{22} & B_{26} & D_{12} & D_{22} & D_{26}\\ B_{16} & B_{26} & B_{66} & D_{16} & D_{26} & D_{66} \end{array}\right]\left\{\begin{array}{c} \begin{array}{c} \varepsilon_{xx0}^{p}\\ \varepsilon_{\theta \theta 0}^{p} \end{array}\\ \gamma_{x\theta 0}^{p}\\ \chi_{xx}^{p}\\ \chi_{\theta \theta }^{p}\\ \chi_{x\theta }^{p} \end{array}\right\}\;\left[\begin{array}{l} Q_{cxx}^{}\\ Q_{c\theta \theta }^{} \end{array}\right]=\kappa_{s}\left[\begin{array}{cc} A_{44} & A_{45}\\ A_{45} & A_{55} \end{array}\right]\left\{\begin{array}{c} \begin{array}{c} \gamma_{\theta r0}^{p}\\ \gamma_{xr0}^{p} \end{array} \end{array}\right\}$$
where $\kappa_{s}$ is the shear correction factor, $A_{ij},B_{ij},D_{ij}$ (*i,j* = 1,2,4,5,6) are tension stiffness, coupled stiffness, and bending stiffness of cylindrical shells, which can be obtained as:(7)$$\left\{A_{ij},B_{ij},D_{ij}\right\}=\sum_{k=1}^{N_{k}}\int_{z_{k}}^{z_{k+1}}{\bar{Q}}_{ij}^{k}\left\{1,z,z^{2}\right\}dz$$
where *N_k_* denotes the number of fiber layers.

#### 2.2.2. For Annular Plate

Using the first order shear deformation theory and taking the cylindrical coordinate system (*o, r, x*) as a base, $\varepsilon_{\gamma }^{0,q}$,$\varepsilon_{\theta }^{0,q}$,$\gamma_{r\theta }^{0,q}$,$\gamma_{rz}^{0,q}$ and $\gamma_{\theta z}^{0,q}$ denote structural strains in the middle surface, $\eta_{r}^{q}$,$\eta_{\theta }^{q}$, and $\eta_{r\theta }^{q}$ are mid-surface change in curvature and twist for the annular plate. They can be written as [17]:(8)$$\begin{array}{l} \varepsilon_{r}^{0,q}=\frac{\partial u_{a}^{q}}{\partial r},\;\;\varepsilon_{\theta }^{0,q}=\frac{u_{a}^{q}}{r}+\frac{1}{r}\frac{\partial v_{a}^{q}}{\partial \theta },\eta_{r}^{q}=\frac{\partial \psi_{r}^{q}}{\partial r},\eta_{\theta }^{q}=\frac{1}{r}\frac{\partial \psi_{\theta }^{q}}{\partial \theta }+\frac{\partial \psi_{r}^{q}}{r},\\ \;\eta_{r\theta }=\frac{1}{r}\frac{\partial \psi_{r}^{q}}{\partial \theta }+\frac{\partial \psi_{\theta }^{q}}{\partial r}-\frac{\psi_{\theta }^{q}}{r},\gamma_{rx}^{0,q}=\psi_{r}^{q}+\frac{\partial w_{a}^{q}}{\partial r},\gamma_{\theta x}^{0,q}=\frac{1}{r}\frac{\partial w_{a}^{q}}{\partial \theta }+\psi_{\theta }^{q},\\ \gamma_{r\theta }^{0,q}=\frac{\partial v_{a}^{q}}{\partial r}+\frac{1}{r}\frac{\partial u_{a}^{q}}{\partial \theta }-\frac{v_{a}^{q}}{r} \end{array}$$

As mentioned above, the force and moment resultants of a thick annular plate can be obtained as:(9)$$\left\{\begin{array}{c} N_{r}\\ N_{\theta }\\ N_{r\theta } \end{array}\right\}=b\int_{-h/2}^{h/2}\left[\begin{array}{c} \sigma_{r}\\ \sigma_{\theta }\\ \tau_{r\theta } \end{array}\right]dz,\left\{\begin{array}{c} M_{r}\\ M_{\theta }\\ M_{r\theta } \end{array}\right\}=b\int_{-h/2}^{h/2}\left[\begin{array}{c} \sigma_{r}\\ \sigma_{\theta }\\ \tau_{r\theta } \end{array}\right]zdz\;\left\{\begin{array}{c} Q_{r}\\ Q_{\theta } \end{array}\right\}=b\int_{-h/2}^{h/2}\left[\begin{array}{c} \tau_{rz}\\ \tau_{\theta z} \end{array}\right]dz$$

### 2.3. Energy Expressions

System Energy includes the energy stored in the cylindrical shells and annular plates, including the corresponding strain energy ($U_{c}^{p}$, $U_{a}^{q}$), which can be written into three parts: tension compression potential energy (*U_S_*), bending potential energy (*U_B_*) and tension compression bending coupling potential energy (*U_BS_*); kinetic energy ($T_{c}^{p}$, $T_{a}^{q}$); potential energy ($U_{cb}^{p}$) stored by the boundary springs of double cylindrical shells and annular plates; and *W_e_* denotes the work done by the imposed external force or moment.

Energy equations above are obtained as follows:(10)$$\begin{array}{l} U_{c}^{}=\frac{1}{2}\int_{S}\{N_{xx}^{}\varepsilon_{xx0}^{}+N_{\theta \theta }^{}\varepsilon_{\theta \theta 0}^{}+N_{x\theta }^{}\gamma_{x\theta 0}^{}+M_{xx}^{}\chi_{xx}^{}\\ \;+M_{\theta \theta }^{}\chi_{\theta \theta }^{}+M_{x\theta }^{}\chi_{x\theta }^{}+Q_{\theta }^{}\gamma_{\theta z0}^{}+Q_{x}^{}\gamma_{xz0}^{}\}dS \end{array}$$
(11)$$U_{S}^{}=\frac{1}{2}\int_{0}^{L}\int_{0}^{2\pi }\left\{\begin{array}{l} A_{11}{\left(\frac{\partial u_{c}}{\partial x}\right)}^{2}+\frac{2A_{12}}{R}(\frac{\partial v_{c}}{\partial \theta }+w_{c})\frac{\partial u_{c}}{\partial x}+\frac{A_{22}}{R^{2}}{\left(\frac{\partial v_{c}}{\partial \theta }+w_{c}\right)}^{2}\\ +2A_{16}(\frac{\partial v_{c}}{\partial x}+\frac{\partial u_{c}}{R\partial \theta })(\frac{\partial u_{c}}{\partial x})+\frac{2A_{26}}{R}(\frac{\partial u_{c}}{R\partial \theta }+\frac{\partial v_{c}}{\partial x})(\frac{\partial v_{c}}{\partial \theta }+w_{c})\\ +A_{66}{\left(\frac{\partial v_{c}}{\partial x}+\frac{\partial u_{c}}{R\partial \theta }\right)}^{2}+\kappa A_{44}{\left(\phi_{c\theta }-\frac{v_{c}}{R}+\frac{\partial w_{c}}{R\partial \theta }\right)}^{2}\\ +\kappa A_{55}{\left(\phi_{cx}+\frac{\partial w_{c}}{\partial x}\right)}^{2}+2\kappa A_{45}(\phi_{c\theta }-\frac{v_{c}}{R}+\frac{\partial w_{c}}{R\partial \theta })(\phi_{cx}+\frac{\partial w_{c}}{\partial x}) \end{array}\right\}Rdxd\theta$$
(12)$$U_{B}^{}=\frac{1}{2}\int_{0}^{L}\int_{0}^{2\pi }\left\{\begin{array}{l} D_{11}{\left(\frac{\partial \phi_{cx}}{\partial x}\right)}^{2}+2\frac{D_{12}}{R}(\frac{\partial \phi_{c\theta }}{\partial \theta })(\frac{\partial \phi_{cx}}{\partial x})+2D_{16}(\frac{\partial \phi_{cx}}{R\partial \theta }+\frac{\partial \phi_{c\theta }}{\partial x})(\frac{\partial \phi_{cx}}{\partial x})\\ +\frac{D_{22}}{R^{2}}{\left(\frac{\partial \phi_{c\theta }}{\partial \theta }\right)}^{2}+2\frac{D_{26}}{R}(\frac{\partial \phi_{cx}}{R\partial \theta }+\frac{\partial \phi_{c\theta }}{\partial x})(\frac{\partial \phi_{c\theta }}{\partial \theta })\\ +D_{66}{\left(\frac{\partial \phi_{cx}}{R\partial \theta }+\frac{\partial \phi_{c\theta }}{\partial x}\right)}^{2} \end{array}\right\}Rdxd\theta$$
(13)$$U_{BS}=\int_{0}^{L}\int_{0}^{2\pi }\left\{\begin{array}{l} B_{11}\frac{\partial u_{c}}{\partial x}\frac{\partial \phi_{cx}}{\partial x}+\frac{B_{12}}{R}\frac{\partial u_{c}}{\partial x}\frac{\partial \phi_{c\theta }}{\partial \theta }+B_{16}\frac{\partial u_{c}}{\partial x}(\frac{\partial \phi_{cx}}{R\partial \theta }+\frac{\partial \phi_{c\theta }}{\partial x})\\ +\frac{B_{12}}{R}(\frac{\partial v_{c}}{\partial \theta }+w_{c})\frac{\partial \phi_{cx}}{\partial x}+\frac{B_{22}}{R^{2}}(\frac{\partial v_{c}}{\partial \theta }+w_{c})\frac{\partial \phi_{c\theta }}{\partial \theta }\\ +\frac{B_{26}}{R}(\frac{\partial v_{c}}{\partial \theta }+w_{c})(\frac{\partial \phi_{cx}}{R\partial \theta }+\frac{\partial \phi_{c\theta }}{R\partial \theta })+B_{16}(\frac{\partial v_{c}}{\partial x}+\frac{\partial u_{c}}{R\partial \theta })\frac{\partial \phi_{cx}}{\partial x}\\ +\frac{B_{26}}{R}(\frac{\partial v_{c}}{\partial x}+\frac{\partial u_{c}}{R\partial \theta })\frac{\partial \phi_{c\theta }}{\partial \theta }+B_{66}(\frac{\partial \phi_{cx}}{R\partial \theta }+\frac{\partial \phi_{c\theta }}{\partial x})(\frac{\partial v_{c}}{\partial x}+\frac{\partial u_{c}}{R\partial \theta }) \end{array}\right\}Rdxd\theta$$
(14)$$T_{c}=\frac{1}{2}\int_{0}^{L}\int_{0}^{2\pi }\left\{\begin{array}{l} I_{0}{\left(\frac{\partial u_{c}}{\partial t}\right)}^{2}+I_{0}{\left(\frac{\partial v_{c}}{\partial t}\right)}^{2}+I_{0}{\left(\frac{\partial w_{c}}{\partial t}\right)}^{2}+2I_{1}\frac{\partial u_{c}}{\partial t}\frac{\partial \phi_{cx}^{}}{\partial t}\\ +2I_{1}\frac{\partial v_{c}}{\partial t}\frac{\partial \phi_{c\theta }^{}}{\partial t}+I_{2}{\left(\frac{\partial \phi_{cx}^{}}{\partial t}\right)}^{2}+I_{2}{\left(\frac{\partial \phi_{c\theta }^{}}{\partial t}\right)}^{2} \end{array}\right\}Rdxd\theta$$
(15)$$U_{cb}=\frac{1}{2}\int_{0}^{2\pi }\left\{\begin{array}{l} {\left[k_{u}u_{c}^{2}+k_{v}v_{c}^{2}+k_{w}w_{c}^{2}+k_{x}\phi_{cx}^{2}+k_{\theta }\phi_{c\theta }^{2}\right]}_{x=0}\\ {\left[k_{u}u_{c}^{2}+k_{v}v_{c}^{2}+k_{w}w_{c}^{2}+k_{x}\phi_{cx}^{2}+k_{\theta }\phi_{c\theta }^{2}\right]}_{x=L} \end{array}\right\}Rd\theta$$
where (*k_u_*, *k_v_*, *k_w_*, *k_x_*, *k**_θ_*) denote the boundary springs related to the boundary constraints of cylindrical shells:(16)$$U_{a}=\frac{1}{2}\int \int \int_{V}\left\{\begin{array}{l} N_{r}\varepsilon_{r}^{0}+N_{\theta }\varepsilon_{\theta }^{0}+N_{r\theta }\gamma_{r\theta }^{0}+M_{r}\eta_{r}\\ +M_{\theta }\eta_{\theta }+M_{r\theta }\eta_{r\theta }+Q_{r}\gamma_{rz}^{0}+Q_{\theta }\gamma_{\theta z}^{0} \end{array}\right\}rdrd\theta dz$$
(17)$$T_{a}=\frac{1}{2}\int_{R_{i}}^{R_{o}}\int_{0}^{2\pi }\left\{\begin{array}{l} I_{0}\left[{\left(\frac{\partial u_{a}}{\partial t}\right)}^{2}+{\left(\frac{\partial v_{a}}{\partial t}\right)}^{2}+{\left(\frac{\partial w_{a}}{\partial t}\right)}^{2}\right]\\ +2I_{1}\left[\left(\frac{\partial u_{a}}{\partial t}\right)\left(\frac{\partial \psi_{r}}{\partial t}\right)+\left(\frac{\partial v_{a}}{\partial t}\right)\left(\frac{\partial \psi_{\theta }}{\partial t}\right)\right]\\ +I_{2}\left[{\left(\frac{\partial \psi_{\theta }}{\partial t}\right)}^{2}+{\left(\frac{\partial \psi_{\theta }}{\partial t}\right)}^{2}\right] \end{array}\right\}rdrd\theta$$
(18)$$U_{ab}=\frac{1}{2}\int_{-\frac{h}{2}}^{\frac{h}{2}}\int_{0}^{2\pi }\left\{\begin{array}{l} R_{o}{\left[k_{ru}u_{a}^{2}+k_{rv}^{}v_{a}^{2}+k_{rw}^{}w_{a}{}^{2}+k_{rx}^{}\psi_{r}^{2}+k_{r\theta }^{}\psi_{\theta }^{2}\right]}_{r=R_{o}}\\ R_{i}{\left[k_{ru}u_{a}^{2}+k_{rv}^{}v_{a}^{2}+k_{rw}^{}w_{a}{}^{2}+k_{rx}^{}\psi_{r}^{2}+k_{r\theta }^{}\psi_{\theta }^{2}\right]}_{r=R_{i}} \end{array}\right\}d\theta dz$$
where (*k_ru_*, *k_rv_*, *k_rw_*, *k_rx_*, *k_r_**_θ_*) denote the boundary springs related to the boundary constraints of an annular plate.

For the coupling conditions between cylindrical shell and annular plate:(19)$${u_{c}^{}|}_{x=p_{a}^{q}}=-w_{a}^{},{v_{c}^{}|}_{x=p_{a}^{q}}=v_{a}^{},{w_{c}^{}|}_{x=p_{a}^{q}}=u_{a}^{},\;{\phi_{c\theta }^{}|}_{x=p_{a}^{q}}=\psi_{\theta }^{}$$

In addition, the potential energy ($U_{ca}$) stored in the coupling springs between cylindrical shell and annular plate is expressed as:(20)$$U_{ca}=\frac{1}{2}\int_{-\frac{h}{2}}^{\frac{h}{2}}\int_{0}^{2\pi }\left\{\begin{array}{l} R_{o}{\left[k_{cu}^{}{\left(u_{c}+w_{a}\right)}^{2}+k_{cv}^{}{\left(v_{c}-v_{a}\right)}^{2}+k_{cw}^{}{\left(w_{c}-u_{a}\right)}^{2}+k_{cx}^{}{\left(\phi_{c\theta }^{}-\psi_{\theta }^{}\right)}^{2}\right]}_{r=R_{o},x=x_{c}}\\ R_{i}{\left[k_{cu}^{}{\left(u_{c}+w_{a}\right)}^{2}+k_{cv}{\left(v_{c}-v_{a}\right)}^{2}+k_{cw}^{}{\left(w_{c}-u_{a}\right)}^{2}+k_{cx}^{}{\left(\phi_{c\theta }^{}-\psi_{\theta }^{}\right)}^{2}\right]}_{r=R_{i},x=x_{c}} \end{array}\right\}d\theta dz$$
where (*k_cu_*, *k_cv_*, *k_cw_*, *k_cx_*) represent the stiffness coefficients of coupling springs.

In order to simplify the calculation, it is assumed that, in the LCDCSS, only the radial point force is acting on the inner and outer shell. In Equation (21), *W_ex_* represents the work done by the external force on the system, δ is the Dirac function, *F*_0_ excitation force amplitude, and the coordinates of the action position:(21)$$W_{ex}=\frac{1}{2}\int_{0}^{L}\int_{0}^{2\pi }F_{0}\delta (x-x_{F})\delta (\theta -\theta_{F})w_{c}^{2}(x,\theta )R_{p}d\theta dx$$

The Lagrangian energy functional (L) of LCDCSS is expressed as:(22)$$L=\sum_{p=1}^{2}\left(T_{c}^{p}-U_{c}^{p}-U_{cb}^{p}\right)+\sum_{q=1}^{Nq}\left(T_{a}^{q}-U_{a}^{q}-U_{ab}^{q}\right)-\sum_{i=1}^{Ni}U_{ca}^{}-W_{ex}$$

*N_i_* is the number of coupling edges.

### 2.4. Displacement Admissible Functions and Solution Process

The modified Fourier series method is used to describe displacement admissible function, in which several supplementary terms are introduced into the Fourier series expansion to remove any potential discontinuities of the displacement and their derivatives throughout the entire solution domain [12,13,40]. The displacement components of the cylindrical shell and annular plate can be defined as:(23)$$\begin{array}{c} U=\sum_{n=-2}^{-1}\sum_{m=-2}^{-1}A\sin\left(\lambda_{\partial m}\partial \right)\sin\left(n\theta \right)+\sum_{m=-2}^{-1}\sum_{n=0}^{N}B\sin\left(\lambda_{\partial m}\partial \right)\cos\left(n\theta \right)+\\ \sum_{n=-2}^{-1}\sum_{m=0}^{M}C\sin\left(\lambda_{\partial m}\partial \right)\cos\left(n\theta \right)+\sum_{m=0}^{M}\sum_{n=0}^{N}D\cos\left(\lambda_{\alpha m}\partial \right)\cos\left(n\theta \right)\\ \partial =c,a \end{array}$$
(24)$$U={\left[u_{c}^{p},v_{c}^{p},w_{c}^{p},\phi_{cx}^{p},\phi_{c\theta }^{p},u_{a}^{q},v_{a}^{q},w_{a}^{q},\psi_{r}^{q},\psi_{\theta }^{q}\right]}^{T}\;A={\left[A_{mn}^{1},B_{mn}^{1},C_{mn}^{1},D_{mn}^{1},E_{mn}^{1},F_{mn}^{1},G_{mn}^{1},H_{mn}^{1},I_{mn}^{1},J_{mn}^{1}\right]}^{T}\;B={\left[A_{mn}^{2},B_{mn}^{2},C_{mn}^{2},D_{mn}^{2},E_{mn}^{2},F_{mn}^{2},G_{mn}^{2},H_{mn}^{2},I_{mn}^{2},J_{mn}^{2}\right]}^{T}\;C={\left[A_{mn}^{3},B_{mn}^{3},C_{mn}^{3},D_{mn}^{3},E_{mn}^{3},F_{mn}^{3},G_{mn}^{3},H_{mn}^{3},I_{mn}^{3},J_{mn}^{3}\right]}^{T}\;D={\left[A_{mn}^{4},B_{mn}^{4},C_{mn}^{4},D_{mn}^{4},E_{mn}^{4},F_{mn}^{4},G_{mn}^{4},H_{mn}^{4},I_{mn}^{4},J_{mn}^{4}\right]}^{T}$$
where **U** is the displacement vectors, and subscripts *c* and *a* represent the cylindrical shell and annular structures in the system, respectively; *λ_am_* = *m*π/*L*, and *λ_bm_* = *m*π/*b,b* = *R_o_* − *R_i_*, denoted the auxiliary polynomial functions introduced to in-plane and anti-plate displacement to remove all the discontinuities potentially associated with the second-order derivatives, then ensuring and accelerating the convergence of the series expansion. A. B, C, and D are the expansion coefficients of the trigonometric series in the displacement allowable functions, and *M* and *N* are the truncated values of the structural displacement function. 

Substitute all the energy formula (Equations (10)–(21)) and the displacement admissible functions (Equations (23) and (24)) of the LCDCSS into the Lagrange energy function (Equation (22)). By taking variation on these equations based on the Rayleigh–Ritz method, characteristic equations of the LCDCSS can be obtained: (25)$$\left\{K-\omega^{2}M\right\}\Pi =F$$

Therefore, the solution process of the vibration characteristics of the system becomes a simple nonlinear equation solution problem. **K** stands for the stiffness matrix and **M** represents the mass matrix separately. **Π** indicates the Fourier coefficients vector, and **F** is the external force contributions, which can be rewritten as vector form *F_s2_* = {*F_s2,u_*,0,0,0,0 }*^T^* and *F_s2_* = {0, *F_s2,v_*,0,0,0}*^T^* when the LCDCSS receive axial and circumferential excitation load. By solving a standard matrix eigenvalue problem, the frequency parameters are then provided.

## 3. Numerical Calculation and Analysis

In this section, the simulation calculation will be carried out for the analysis model of the LCDCSS. Firstly, the convergence property and calculation accuracy are verified to show the correctness and effectiveness of the structure model. Then, the parametric investigations of free vibration characteristics of the LCDCSS are carried out, covering the coupled situation and constraint condition of the annular plate, boundary conditions, geometric features, and so on. After that, forced vibration characteristics of the LCDCSS are studied, and some new results and new laws are obtained to enrich the research field.

### 3.1. Convergence and Validation Study of the LCDCSS

The verification of convergence is of great significance to guarantee the calculation accuracy. In the Table 1 calculation example, three annular plates coupled with cylindrical shells rigidly, which distribute in the axial direction of the cylindrical shell as P_a_ = [1/12,1/2,11/12] L. The material parameters of the inner and outer cylindrical shells are E_1_ = 50 GPa, E_2_ = 2 GPa, G_12_ = G_13_ = 1 GPa, G_23_ = 0.4 GPa, μ = 0.25, and the laying scheme is $\alpha_{\mathrm{fiber}}$ = [0° 90° 0°]. Furthermore, the physical properties of annular plates are as E_1_ = 150 GPa, E_2_ = 10 GPa, G_12_ = G_13_ = 6 GPa, G_23_ = 5 GPa, μ = 0.25, $\alpha_{\mathrm{fiber}}$ = [0° 90° 0° 90°], ρ = 1500 kg/m^3^. Moreover, geometric parameters of the system are as: L = 1.2 m, R_o_ = 0.5 m, R_i_ = 0.4 m, h_c_ = h_a_ = 0.005 m. For the description of the boundary conditions, the symbols F, C, S, and SD to represent the free boundary, the clamped boundary, the simply supported boundary, and shear diaphragm supported boundary, respectively. For example, the boundary conditions of LCDCSS, written with the form of FF-CC denoting ends with x_p_ = 0 of the double cylindrical shell, are F, while ends with x_p_ = L of the double cylindrical shell are C. According to the research on rotary composite structures in reference [10], in the calculation later, the stiffness value of coupling spring remains 1 × 10^14^ N/m to stimulate a rigid connection effect.

To verify the convergence, the truncated number M and N of substructures are chosen as the same value, which range between 6–48. Table 1 records the lowest five frequency parameters of an LCDCSS under different classical boundary conditions. The numerical comparison shows that natural frequency is obviously changed when M ranges from 6 to 18, and the maximum deviation is 4.82%. The frequency deviation decreases significantly; when M changes from 18 to 26, the maximum deviation decreases to 1.11%, and the tendency flattens out when M over 32 shows that the calculation results tend to converge. The truncated numbers are defined as constant in this regard, which can satisfy the calculation precision.

Next, more examples of the analytical model are analyzed to test the validity and accuracy. Due to vibration characteristics of a laminate composite, a double cylindrical shell structure has not been published. Jin et al. [27] studied the vibration characteristics of an annular plate coupled double-layer cylindrical shell system (DCSS) of isotropic materials, providing reference data as well as the finite element method.

The material parameters of annular plate, inner shell, and outer shell in the system are the same as: *E*_1_ = *E*_2_ = 206 GPa, *G*_12_ = *G*_13_ = *G*_23_ = *E*_1_/2(1 + *μ*), *μ* = 0.3, ρ = 7850 kg/m^3^, and geometric parameters are: *L* = 1.2 m, *R_o_* = 0.5 m, *R_i_* = 0.4 m, *h_c_* = *h_a_* = 0.005 m. The results of the theoretical model under different boundary conditions are compared with those of reference and finite element calculations, which are analyzed by ABAQUS software. The element shell 181 is selected for finite element calculation, and the element edge length is set to 0.01.

As shown in Table 2, the system with three annular plates coupled between cylindrical shells is investigated. The annular plates are located at *p*_a_ = [1/12,1/2,11/12] *L,* respectively. The boundary conditions are chosen as a different classical boundary, and the annular plates are rigidly coupled with the inner and outer cylindrical shells. It can be seen from the data comparison that the largest error between the present and Ref [27] is lower than 3%. The present results are in good agreement with FEM. On the whole, the calculation results of this method are consistent with Ref and FEM. The error of the two methods does not increase significantly as the order of mode increases. The proposed method maintains reliable calculation accuracy.

Figure 2 shows some modes in Table 2. It can be seen from the figure that, under three different classical boundary conditions, the vibration modes of the inner and outer cylindrical shell in DCSS system are always consistent, indicating that the energy transfer between substructures is relatively stable, and the system has high stability. It is verified that the proposed method can predict the natural vibration characteristics of DCSS accurately. Then, parametrical study on free vibration of the LCDCSS is carried out.

### 3.2. Free Vibration Analysis of the LCDSS

Firstly, the influence of the coupling relationship between the shells and the annular plates on the vibration characteristics is investigated conveniently. In the next numerical example, there are two annular plates between double cylindrical shells, which are located at p_a_ = [1/3 2/3] L. One of four sets spring on coupled boundaries is set to 10^5^ maintaining others’ rigidity. The physical parameters of cylindrical shell and annular plates are still consistent with Figure 2, and the geometric parameters of the LCDCSS are defined as: $\alpha_{\mathrm{fiber}}$= [0° 90° 90° 0°], [45° −45° 45° −45°], L = 3, R_o_ = 0.5 m, R_i_ = 0.3 m, 1/4h_a_ = h_c_ = 0.005 m.

From calculations in Table 3, it can be seen that the radial spring stiffness k_cu_ has an obvious influence on the natural frequency of the structure, which shows that the out-plane vibration of an annular plate has a great influence on the natural frequency of the coupling system, especially on the lower order frequency. Moreover, according to the comparison of the calculations with the two laying schemes, the natural vibration frequency of $\alpha_{\mathrm{fiber}}$ = [45° −45° 45° −45°] is higher than that of [0° 90° 90° 0°].

Then, the influence of coupled position of annular plate on structural vibration is investigated. There is annular plate coupling in the system, and the position moves from *x_c_* = 0 to *x_c_* = l end along the cylindrical shell axis. In the calculation, the annular plate and cylindrical shells couple rigidly. The dimensionless frequency is Ω = ω*L^2^*/*h* (*ρ*/*E*_2_)^1/2^. Figure 3 shows the free vibration frequency variation curves of different orders when the annular plate moves along the axial direction of the cylindrical shell under the fixed and simply supported boundary conditions. Frequency parameters in Figure 3 are distributed symmetrically on the left and right, which reflect the axial symmetry of double cylindrical shell structure.

Figure 4 investigates change of LCDCSS free vibration frequency caused by different fiber angles under classical and elastic boundary conditions. Coupling position and material parameters of the system are consistent with Table 3, and the geometric parameters are as: *L* = 3 m, *R_o_* = 0.5 m, *R_i_* = 0.3 m, *h_c_*¼ = *h_a_* = 0.005 m. The laying scheme of laminated material is [90° $\alpha_{\mathrm{fiber}}$ 90°], in which $\alpha_{\mathrm{fiber}}$ changes from 0 to 180. It can be found in Figure 5 that curves of natural frequencies are symmetrically distributed, and there is an extremum value at the midpoint where $\alpha_{\mathrm{fiber}}$ = 90°. It can be concluded from the results that, when the angle between adjacent fibers is 0°, the natural vibration frequency of the structure is the minimum.

The model we have established is abstracted from the actual structure of the project, which needs to consider the specific requirements of the actual working conditions for the structural dynamic characteristics. Theoretically, the greater the number of annular plates as the inner and outer shell connection structure, the higher the stability of the system. However, it can be seen from the calculation results given in the figure that the length of the cylindrical shell is l = 0.5 m, and other geometric and material parameters are the same as those in Table 1. As shown in Figure 5, for a double cylindrical shell with finite length, when the number of annular plates increases from 1 to 10 and distributes along the *x*-axis direction of the cylindrical shell, the structural frequency parameters first increase significantly. However, when the number of annular plates exceeds 6, the structural frequency parameters tend to be stable, and the impact of continuous increase of annular plates on the structural frequency is reduced. It can be estimated that the minimum number of annular plates can be used to ensure the stability of the LCDCSS.

In the example shown in Figure 6, the influence of the thickness coefficient *h_o_*/*h_i_* of inner and outer cylindrical shells on the vibration frequency of LCDCSS is studied. The system parameters are set as: *L* = 1.5 m, *h_i_* = *L*/500, *h_o_*/*h_i_* = 1~20, *R_i_* = 0.3 m, *R_o_* = 0.5 m, *E*_1_ = 740 GPa, *E*_2_ = 18.5 GPa, *μ*_12_ = 0.25, *G*_12_ = *G*_13_ = 111 GPa, *G*_23_ = 92.5 GPa, $\alpha_{\mathrm{fiber}}$ = [30° 0° −30° 0° 30°], *ρ* = 1600 kg/m^3^. As showed in Figure 6, with the increase of *h_o_*/*h_i_*, the variation extent of low-order vibration frequency is small, and the high-order frequency increases significantly. When the thickness coefficient *h_o_*/*h_i_* increases to a certain threshold, the variable trend of the system with classical boundary conditions listed in the figure decreases obviously.

Figure 7 shows the variation of vibration characteristics of LCDCSS when the thickness coefficient of both cylindrical shell and annular plates changes simultaneously. The system parameters are as: $\alpha_{\mathrm{fiber}}$ = [0° 90° 0° 90°], *L =* 2 m, *h_a_* = 0.02~0.4, *h_i_* = 0.004 m, *h_o_/h_i_* = 1~20. From Figure 7, it can be found that the frequencies continue to rise with the increase of structural thickness, which shows that the larger the thickness, the system is more stable. It shows that the geometric parameters affect the vibration characteristic stiffness matrix of the LCDCSS in a certain range, and the vibration characteristics of the system can be adjusted by changing the parameters, which has important practical significance for the performance control of double shell structure.

### 3.3. Steady State Response Analysis of the LCDCSS 

This section attempts to verify the effectiveness of the method proposed to predict forced vibration characteristics of the LCDCSS. Firstly, take the isotropic material double-cylindrical shell (DCSS) as an example to test the effectiveness of the model for the steady state response analysis. When the boundary condition of FF-CC is considered, there are two annular plates coupled in DCSS located at *P_a_* = [0 1/2]*L*. The physical and geometric properties of the DCSS are defined as: *E* = 206 GPa, *G*_12_ = *G*_13_ = *G*_23_ = *E*/2 (1 + *μ*), *μ* = 0.3, *ρ* = 7850 kg/m^3^, *L* = 1.2 m,*R_o_* = 0.5 m, *R_i_* = 0.4 m, *h_c_* = *h_a_* = 0.005 m. It is assumed that the outer cylindrical shell of the double-layer system is subjected to a point force *F* = 1 N located at (*x* = 0.6 m, *θ* = 60°) in cylindrical coordinate (*o*, *x*, *θ*, *r*), which is opposite to the normal direction. Results of measure point 1# located at (*x* = 0.3 m, *θ* = 60°) on the outer shell, point 2# located at (*x* = 0.3 m, *θ* = 60°) on the inner shell, point 3# located at (*r* = 0.05, *θ* = 60°) on the annular plate on the left end of the cylindrical shell. As shown in Figure 8, the out-plane response curves of the measure points with this method are compared with the FEM results. In the finite element calculation, 181 shell elements are selected as calculation units, mesh size is 0.01 × 0.01 mm, and its results coincide quite well with the results mentioned above, which proves that a parametrical study on the forced vibration of the system will be carried out accurately and reliably.

As Figure 9 shows, variation of response vibration frequency of LCDCSS with various layer scheming is investigated. The material properties of the cylindrical shells are as follows: *E*_1_ = 50 GPa, *E*_2_ = 2 GPa, *μ*_12_ = 0.25, *G*_12_ = *G*_13_ = 1 GPa, *G*_23_ = 0.4 GPa, *ρ* = 1600 kg/m^3^, and those of annular plates are: *E*_1_ = 150 GPa, *E*_2_ = 10 GPa, *μ*_12_ = 0.25, *G*_12_ = *G*_13_ = 6 GPa, *G*_23_ = 5 GPa, *ρ* = 1600 kg/m^3^. The laying scheme of cylindrical shell and annular plate is $\alpha_{\mathrm{fiber}}$ = [0° 90° 0° 90°]*_n_* (*n* = 2, 3, 4). In addition, the geometric parameters, boundary conditions, position and size of external force excitation of LCDCSS are consistent with Figure 8. Results of measure points are: point 1# located at (*x* = 0.3 m, *θ* = 60°) on the outer shell, point 2# located at (*x* = 0.3 m, *θ* = 60°) in the outer shell, point 3# located at (*r* = 0.05, *θ* = 60°) on the annular plate coupled on the left end of cylindrical shell, point 4# located at (*r* = 0.05, *θ* = 90°) coupled in the middle of the cylindrical shell. The curve comparison shows that, with the increase of the number of fiber layers, the response amplitude changes slightly, but the change amplitude is small. The formant of the response displacement curve moves to the right, which is more obvious in the higher frequency region. Such characterization shows that the increase of the fiber layer can improve the structural stability.

In Figure 10, the influence of excitation amplitude on response vibration is investigated. In the LCDCSS system, an annular plate couples in the axial middle of the cylindrical shell. The boundary condition and laying scheme is CC-CC with other parameters including external force and measure points are consistent with Figure 8. From the comparison of the response curves of F = 1N, 2N, 3N in Figure 10, it can be seen that the peak response increases in proportion to the increase of the corresponding excitation amplitude. However, waveform of the response curve of the system will not change, and the displacement curve will not shift within the frequency range, which is consistent with the general cognition.

Figure 11 shows the axial vibration response frequency of LCDCSS at four response points with thickness of the annular plate is *h_a_* = 0.007 m, 0.008 m, and 0.009 m, respectively. From the change of the response curves in Figure 11, it can be seen that the formant in the frequency region moves to the right, and the frequency amplitude also decreases, which means that the structural stiffness is improved. The greater the thickness of annular plate, the stronger the structural constraints.

## 4. Conclusions

Based on the Rayleigh–Ritz energy method, a unified analysis model for the vibration characteristics of the LCDCSS with general boundary condition is established in this paper. The accuracy and reliability of the method are verified by simulation calculation of different examples and compared with the results of literature and finite element method. Through the parametric study, free vibration and forced vibration of double cylindrical shell system are predicted, which enriches the relevant data in the research field and provides reference for the structure design. The research results show that the method can effectively solve the problems of free vibration and forced vibration of LCDCSS with an arbitrary annular plate. This method can be further extended to the study of vibration characteristics of multilayer plates, multilayer shells, and more complex coupled structural systems.

## Figures and Tables

**Figure 1 materials-15-04246-f001:**
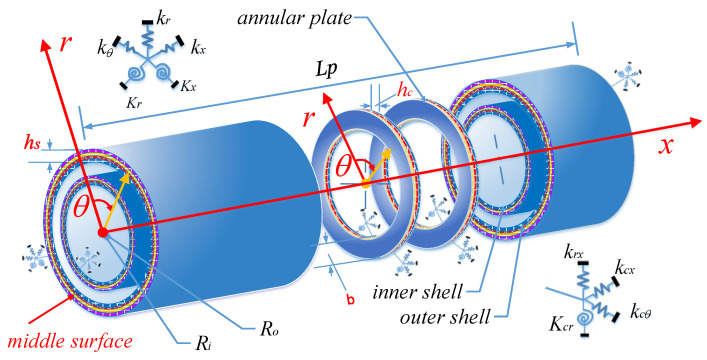
Model of the laminated composite double cylindrical shell system (LCDCSS) with general boundary conditions.

**Figure 2 materials-15-04246-f002:**
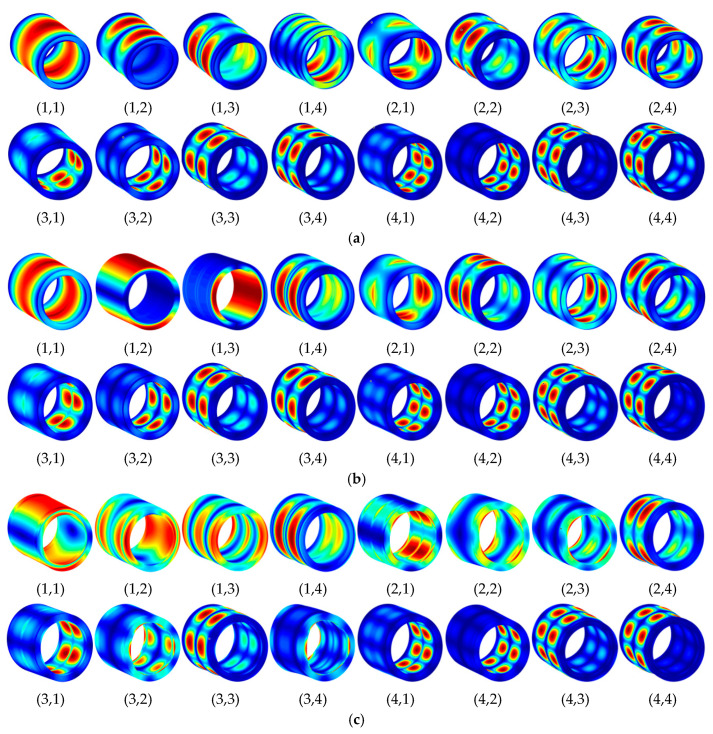
Mode shapes of DCSS with different boundary conditions. (**a**) CC-CC; (**b**) SS-SS; (**c**) FF-FF.

**Figure 3 materials-15-04246-f003:**
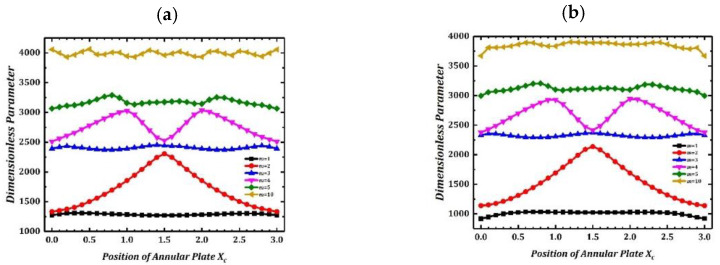
Variation of frequency parameters of LCDCSS with various coupling positions. (**a**) CC-CC; (**b**) SS-SS.

**Figure 4 materials-15-04246-f004:**
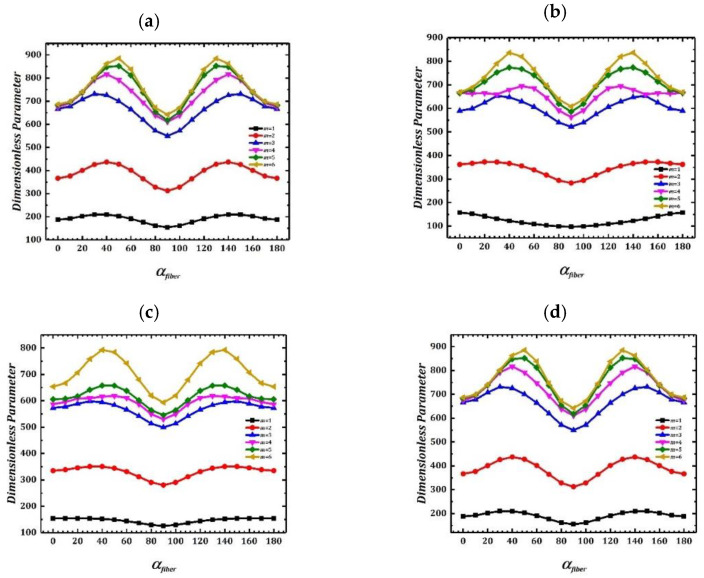
Variation of frequency parameter with various fiber angles of LCDCSS under classical and elastic boundary conditions. (**a**) CC-CC; (**b**) SS-SS; (**c**) E_1_E_1_-E_1_E_1_; (**d**) E_2_E_2_-E_2_E_2_.

**Figure 5 materials-15-04246-f005:**
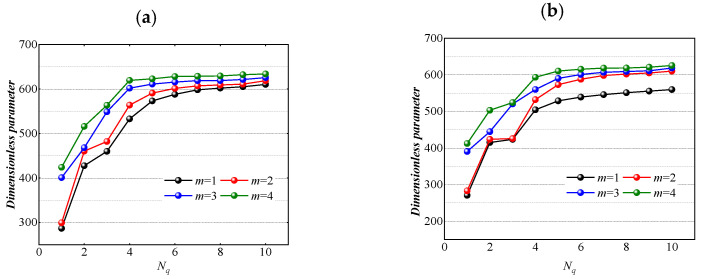
Variation of frequency parameter of LCDCSS with a various number of annular plates. (**a**) CC-CC; (**b**) SS-SS.

**Figure 6 materials-15-04246-f006:**
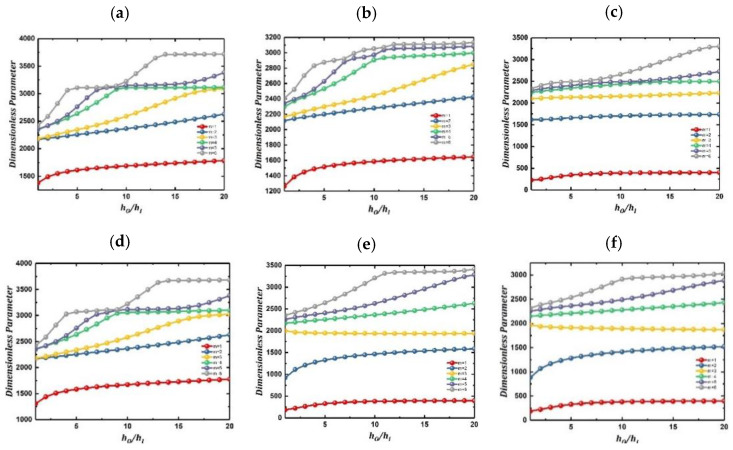
Variation of frequency parameter Ω versus thickness for an LCDCSS with classical boundary conditions. (**a**) CC-CC; (**b**) SS-SS; (**c**) FF-FF; (**d**) CS-CS; (**e**) CF-CF; (**f**) SF-SF.

**Figure 7 materials-15-04246-f007:**
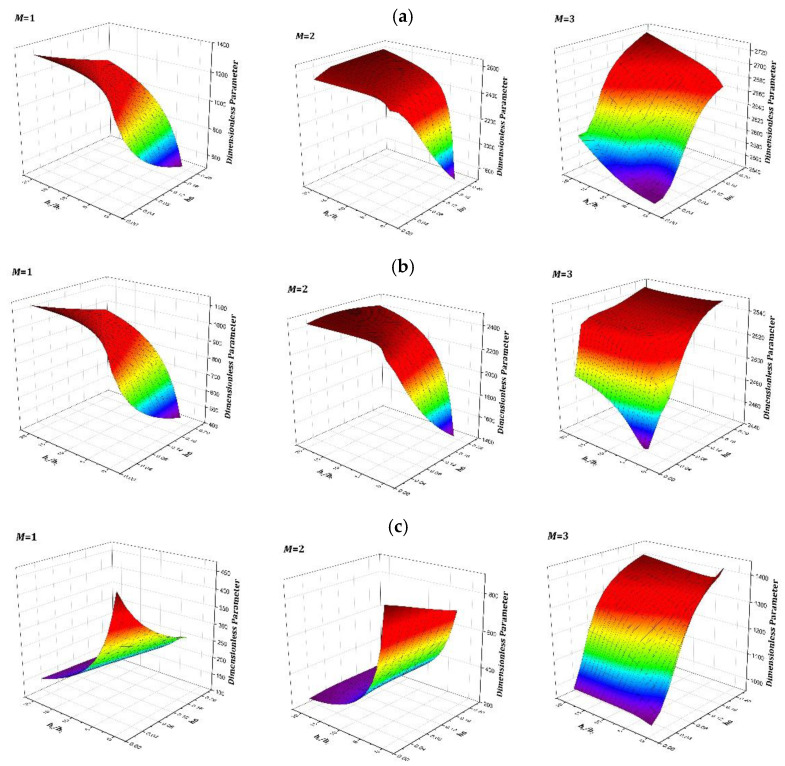
Variation of frequency parameter with various thickness of cylindrical shell and annular plate in LCDCSS. (**a**) CC-CC; (**b**) SS-SS; (**c**) E_1_E_1_-E_1_E_1_.

**Figure 8 materials-15-04246-f008:**
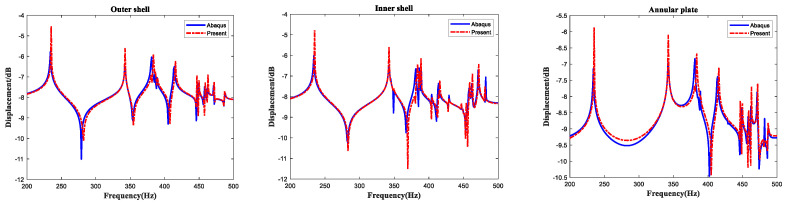
Comparison of steady state response of DCSS.

**Figure 9 materials-15-04246-f009:**
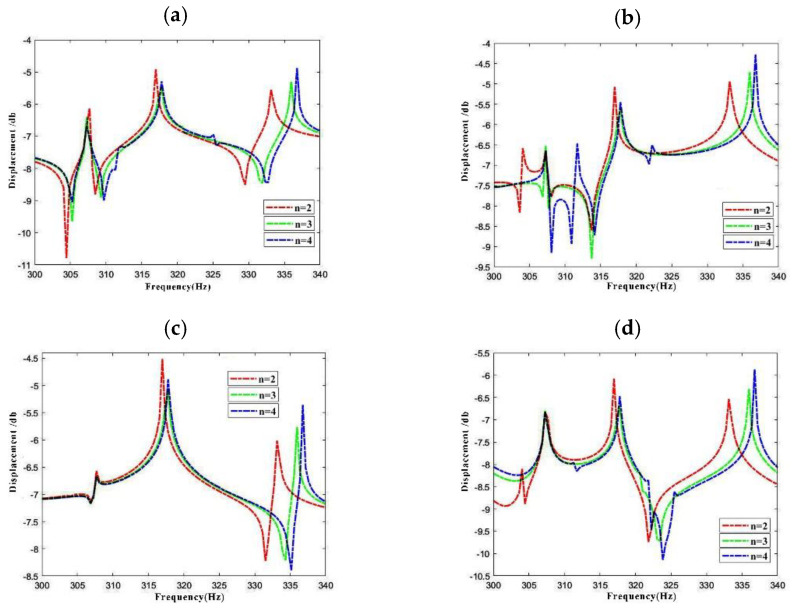
Variation of response vibration frequency for LCDCSS with various material layer scheming. (**a**) Point 1#; (**b**) Point 2#; (**c**) Point 3#; (**d**) Point 4#.

**Figure 10 materials-15-04246-f010:**
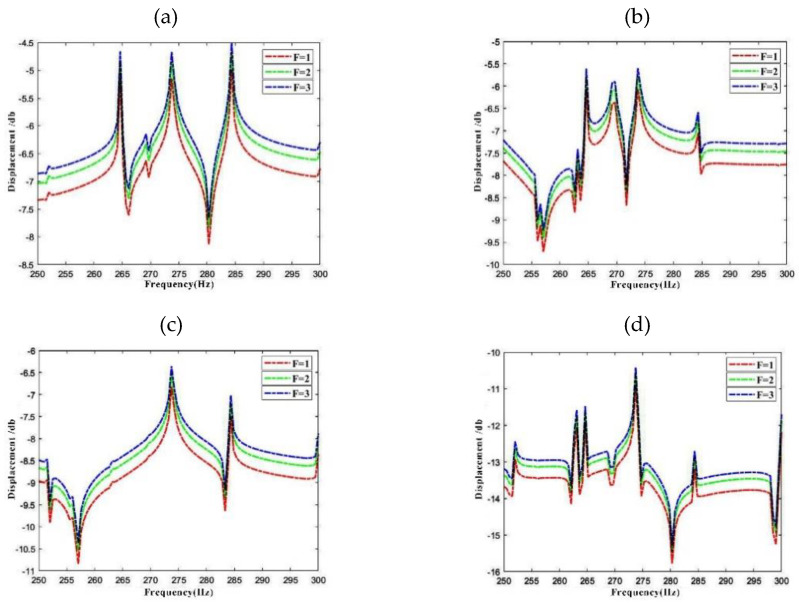
Variation of response vibration frequency for LCDCSS with various external excitation amplitude. (**a**) Point 1#; (**b**) Point 2#; (**c**) Point 3#; (**d**) Point 4#.

**Figure 11 materials-15-04246-f011:**
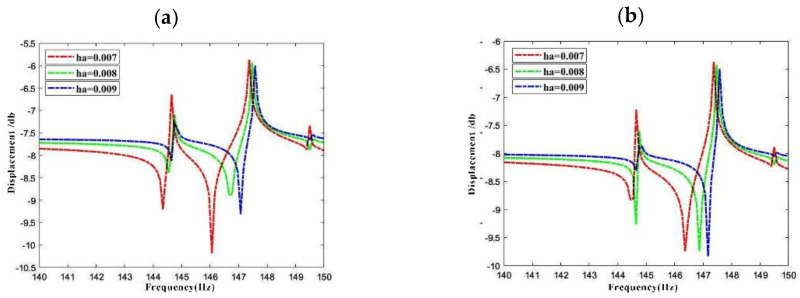
Variation of axial dynamic response vibration frequency for LCDCSS with various thicknesses of annular plates. (**a**) Point 1#; (**b**) Point 2#; (**c**) Point 3#; (**d**) Point 4#.

**Table 1 materials-15-04246-t001:** The lowest five frequency parameters of an LCDCSS with various truncation values.

	*n* = 3												
CC -CC	** *m* **	**M = 6**	**M = 10**	**M = 14**	**M = 18**	**M = 22**	**M = 26**	**M = 30**	**M = 34**	**M = 38**	**M = 42**	**M = 46**	**M = 50**
	1	206.2	201.8	198.8	196.2	195.9	195.0	194.5	194.2	193.9	193.8	193.6	193.4
	2	254.8	253.7	251.6	250.1	249.3	248.7	248.5	248.4	248.1	247.9	247.8	247.6
	3	285.1	272.4	267.4	266.4	266.1	265.8	265.7	265.6	265.5	265.4	265.4	265.3
	4	295.7	283.5	278.1	275.1	275.0	274.3	274.0	273.8	273.6	273.6	273.5	273.4
	5	408.2	401.2	397.1	394.4	393.4	392.2	391.7	391.4	390.8	390.5	390.3	390.1
FF -FF	1	190.2	185.0	181.7	180.3	179.4	178.1	177.6	177.2	176.8	176.7	176.5	176.3
	2	220.1	217.6	212.8	210.9	210.8	208.6	208.1	207.9	207.3	207.0	206.9	206.6
	3	270.1	268.4	266.0	265.4	265.4	264.8	264.7	264.7	264.6	264.5	264.5	264.4
	4	285.8	278.4	274.6	272.7	271.9	271.2	270.9	270.7	270.6	270.5	270.4	270.3
	5	333.6	328.6	322.8	320.2	320.1	316.9	316.5	316.1	315.3	315.0	314.7	314.3
SS -SS	1	200.4	195.8	192.7	191.1	190.4	189.2	188.7	188.3	188.0	187.9	187.8	187.6
	2	252.3	249.6	246.9	245.7	245.7	244.4	244.2	243.8	243.6	243.6	243.5	243.3
	3	272.3	270.0	266.9	266.1	266.0	265.4	265.3	265.1	265.1	265.1	265.0	264.9
	4	288.3	280.1	275.7	273.7	272.9	272.0	271.7	271.5	271.3	271.3	271.2	271.1
	5	400.2	392.7	388.3	386.2	386.0	384.0	383.5	382.7	382.4	382.3	382.1	381.8
FC -FC	1	201.6	196.6	196.6	191.9	191.2	190.0	189.5	189.1	188.8	188.7	188.5	188.3
	2	246.3	244.6	244.6	240.7	240.6	239.0	238.8	238.6	238.3	238.1	238.0	237.8
	3	273.2	269.5	269.5	265.7	265.6	265.1	265.0	264.9	264.8	264.8	264.7	264.7
	4	287.9	280.0	280.0	274.1	273.3	272.5	272.2	272.1	271.9	271.8	271.7	271.6
	5	390.4	381.2	381.2	373.3	373.0	370.5	370.1	369.7	369.1	368.8	368.5	368.2

**Table 2 materials-15-04246-t002:** The first four natural frequencies of DCSS coupled with three annular plates.

BC	*m*	*n* = 1			Error 1	Error 2	*n* = 2			Error 1	Error 2
		Ref [27]	FEM	Present	(%)	(%)	Ref [27]	FEM	Present	(%)	(%)
CC –CC	1	830.67	830.84	830.78	−0.013	0.008	540.78	540.57	541.40	−0.114	−0.153
	2	1382.7	1377.3	1382.6	0.005	−0.387	979.84	975.19	980.62	−0.080	−0.557
	3	1385.5	1382.3	1383.8	0.125	−0.106	995.01	996.17	995.89	−0.089	0.028
	4	1536.2	1534.1	1534.5	0.109	−0.028	1083.9	1083.3	1083.7	0.022	-0.033
SS –SS	1	773.99	774.31	774.33	−0.044	−0.003	428.31	427.85	429.25	−0.220	−0.328
	2	974.97	973.03	974.92	0.005	−0.194	950.53	944.75	951.24	−0.075	−0.687
	3	1218.2	1215.6	1217.6	0.046	−0.168	972.67	974.36	974.13	−0.150	0.024
	4	1380.1	1372.6	1379.6	0.036	−0.510	1076.4	1075.4	1076.0	0.034	−0.059
FF –FF	1	956.58	950.88	957.90	−0.138	−0.738	210.99	208.23	211.88	−0.424	−1.755
	2	1227.3	1221.2	1224.5	0.229	−0.269	236.92	235.29	240.68	−1.589	−2.293
	3	1271.1	1271.1	1272.1	−0.078	−0.078	777.66	777.80	779.93	−0.292	−0.274
	4	1355.3	1346.4	1355.7	−0.028	−0.689	952.73	950.88	946.29	0.676	0.483
BC	*m*	*n* = 3			Error 1	Error 2	*n* = 4			Error 1	Error 2
		Ref [27]	FEM	Present	(%)	(%)	Ref [27]	FEM	Present	(%)	(%)
CC –CC	1	523.93	517.91	521.89	0.389	−0.769	545.06	543.67	546.61	−0.284	−0.540
	2	740.85	742.00	742.18	−0.179	−0.024	588.35	587.24	590.35	−0.339	−0.529
	3	761.54	756.21	759.44	0.275	−0.428	626.64	626.09	625.57	0.171	0.084
	4	805.56	804.71	804.63	0.115	0.010	627.56	626.57	627.94	−0.060	−0.218
SS –SS	1	445.11	443.01	448.10	−0.672	−1.150	502.42	501.33	505.14	−0.542	−0.761
	2	687.93	689.70	690.68	−0.400	−0.142	541.87	545.29	546.62	−0.876	−0.243
	3	722.33	718.60	722.60	−0.038	−0.557	596.94	594.84	596.85	0.015	−0.338
	4	785.97	785.31	785.31	0.083	0.000	601.94	600.24	600.34	0.266	−0.017
FF –FF	1	429.45	425.74	431.62	−0.506	−1.382	500.47	503.97	503.74	−0.653	0.046
	2	515.85	510.74	520.01	−0.807	−1.816	541.04	542.63	544.53	−0.644	−0.349
	3	720.87	717.15	721.64	−0.107	−0.626	590.51	588.05	589.81	0.119	−0.299
	4	729.39	723.6	733.88	−0.615	−1.420	599.67	597.89	597.77	0.317	0.020

**Table 3 materials-15-04246-t003:** Influence of coupling relationship on vibration frequency of LCDCSS.

	[0° 90° 90° 0°]				[45° −45° 45° −45°]			
B.C.	CC–CC				CC–CC			
*m*	*k_cu_ = 10^5^*	*k_cv_ = 10^5^*	*k_cw_ = 10^5^*	*k_cx_ = 10^5^*	*k_cu_ = 10^5^*	*k_cv_ = 10^5^*	*k_cw_ = 10^5^*	*k_cx_ = 10^5^*
1	44.967	189.840	191.196	190.676	57.105	282.229	305.843	309.305
2	44.971	360.911	372.063	371.453	57.213	482.443	657.440	682.661
3	190.583	553.187	671.764	674.137	311.144	494.711	1029.792	999.627
4	371.525	629.980	679.502	678.936	689.441	677.832	1099.645	1053.731
5	674.644	687.826	688.259	684.037	745.276	832.207	1132.119	1144.327
B.C.	SS–SS				SS–SS			
*m*	*k_cu_ = 10^5^*	*k_cv_ = 10^5^*	*k_cw_ = 10^5^*	*k_cx_ = 10^5^*	*k_cu_ = 10^5^*	*k_cv_ = 10^5^*	*k_cw_ = 10^5^*	*k_cx_ = 10^5^*
1	44.96	165.65	166.72	166.196	57.08	163.09	161.20	166.102
2	44.97	357.85	368.91	368.460	57.20	374.63	543.35	565.734
3	166.14	538.79	607.20	591.192	166.77	431.90	969.53	953.477
4	368.32	606.6	660.2	667.569	571.7	658.95	1093.0	1052.548
5	636.8	625.42	673.8	674.025	744.76	698.5	1125.9	1141.095
B.C.	FF–FF				FF–FF			
*m*	*k_cu_ = 10^5^*	*k_cv_ = 10^5^*	*k_cw_ = 10^5^*	*k_cx_ = 10^5^*	*k_cu_ = 10^5^*	*k_cv_ = 10^5^*	*k_cw_ = 10^5^*	*k_cx_ = 10^5^*
1	44.968	202.37	222.30	200.59	57.079	178.34	245.07	246.23
2	45.515	282.73	297.49	294.36	57.888	238.50	282.33	289.75
3	195.49	378.91	385.95	384.23	242.777	331.24	355.58	367.23
4	294.88	452.46	472.03	470.0	287.619	424.28	685.21	718.05
5	384.34	573.59	659.19	646.0	373.156	516.01	1046.3	1015.0

## Data Availability

All data, models, and code generated or used during the study appear in the submitted article.

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
