# Peer review of "Vibration Characteristics of a Laminated Composite Double-Cylindrical Shell System Coupled with a Variable Number of Annular Plates"

_materials, 2022, doi:10.3390/ma15124246_

Round 1

Reviewer 1 Report

In this paper, The authors consider a unified analytical model for the vibration characteristics of laminated double cylindrical shells coupling with variable number of annular plates. Artificial virtual boundary as well as the virtual coupling spring techniques are used to simulate the constraint relationship. The unknown coefficients of the displacement components are obtanied by making use of the Rayleigh-Ritz method.  The subject of the paper is interesting and in general the paper is well-written. The reference list is convenient. I recommend a minor revision according to the following comments:

1. In the Introduction Section, the first paragraph and the second one are exactly the same. Therefore, one of them should be erased.

2. After the phrase "Based on shear shell theory (SDST)" on Page 4, Line 161, a reference should be cited. Similarly, another reference should be cited after the phrase "According to the linear elasticity theory" on Page 4, Line 169.

3. A future problem should be provided in Section 4.

4. The paper should be language checked. There are typos in the paper.

Author Response

Thank you very much for your serious and responsible attitude. This comment is valuable in helping readers understand the content of this paper and in improving the quality of our manuscript. We have tried our best to improve the manuscript. Please see the attachment.

Reviewer 2 Report

The subject of the paper might be interesting but there are many problems in the writing of the manuscript which prevent to publish it. It seems that the authors paid no attention to some basic checking about the references, final version of the paper, etc.

1.- Lines 28 to 38 are copied in lines 39 to 49 in the Introduction

2.- All the references mentioned in the paper do not match with the references listed in the reference list. For instance, Ming et al [1] reference is Altekin, Lee and Ready [2] reference is Gorman, and so on. (all the references are wrong).

It seems that either, the text of the introduction corresponds to a different paper or the list of published works corresponds to a different paper.

This discrepancy makes the whole section 1 of the paper invalid.

3.- Line 176 is wrong:  TAUp cxx, does not appear in formula (4). In fact, for a plane stress assumption for a single monolayer, only direct stresses and shear streeses appear. Bending stress makes no sense.

4.- Line 190 repeats the title and section number 2.2.1 For cylindrical shell that appears in line 160.

5.- Equation (20) is missing, between lines 213 and 214

6.- Line 216, equation (25) is mentioned, but is does not appear. Perhaps equation (21)?

7.- Line 221, section 2.3 appears, but the section number is already used inline 197

8.- Lines 235 and 236 are wring, as they refer, for instance to the displacement admissible function (equation) while that equation corresponds to the Lagrangian energy functional, and not equation (26)

9.- Line 254, the acronym LCDPS appears for the first and only time in the paper, probably is wrong. Otherwise, the authors should explain the meaning.

10.-Line 304. It appears LCDSS, for the first time. Probably is LCDCSS

11.- Line 327 refers to figure 6, but I think it should be figure 5

12..- Line 352 repeats the number section 3.2, that also appears at line 304

Author Response

Thank you very much for your serious and responsible attitude. This comment is valuable in helping readers understand the content of this paper and in improving the quality of our manuscript. We have tried our best to improve the manuscript. We have carefully reviewed the entire paper and revised the incorrect or irregular wording and language in the paper.

Reviewer 3 Report

This work investigates vibration characteristics of laminated composite double-cylindrical shell system coupled with variable number of annular plates. This work falls in the scope of the journal. Some new results of laminated composite double-cylindrical shell system coupled with variable number of annular plates have been proposed. However, some issues should be addressed as follows:

1.    Firstly, please mention in the manuscript the difference between this work and the previously published one (https://doi.org/10.1016/j.compstruct.2021.115020). These two works look similar

2.    Can we use different theories for the shell system and the annular plates? For example, for the shell system we use first order shear deformation theory but for the annular plates we use the third order shear deformation theory

3.    Can you estimate the minimum number of annular plates to ensure the coupled system can work well?

4.    It seems that Authors do not define the notations in the text such as: CC-CC, SS-SS, FF-FF, etc.

5.    Many typing mistakes were found such as: “::” (line 175, 186); “2.2.1. For cylindrical shell” line 190, etc. Please revise the manuscript. Also, the last section “4. Discussions” I think that you should change to “4. Conclusions”

Author Response

Thank you very much for your serious and responsible attitude. This comment is valuable in helping readers understand the content of this paper and in improving the quality of our manuscript. We have tried our best to improve the manuscript. We have carefully reviewed the entire paper and revised the incorrect or irregular wording and language in the paper. The following contents have been added to the revised manuscript.

Round 2

Reviewer 2 Report

The paper is ready to be published